# Fibroblasts and Endothelial Cells in Three-Dimensional Models: A New Tool for Addressing the Pathogenesis of Systemic Sclerosis as a Prototype of Fibrotic Vasculopathies

**DOI:** 10.3390/ijms25052780

**Published:** 2024-02-28

**Authors:** Caterina Bodio, Alessandra Milesi, Paola Adele Lonati, Cecilia Beatrice Chighizola, Alessandro Mauro, Luca Guglielmo Pradotto, Pier Luigi Meroni, Maria Orietta Borghi, Elena Raschi

**Affiliations:** 1Experimental Laboratory of Immunological and Rheumatologic Researches, Istituto di Ricovero e Cura a Carattere Scientifico (IRCCS), Istituto Auxologico Italiano, 20095 Cusano Milanino, Italy; c.bodio@auxologico.it (C.B.); p.lonati@auxologico.it (P.A.L.); p.meroni@auxologico.it (P.L.M.); or maria.borghi@unimi.it (M.O.B.); 2Laboratory of Clinical Neurobiology, Istituto di Ricovero e Cura a Carattere Scientifico (IRCCS), Istituto Auxologico Italiano, 28824 Piancavallo, Italy; a.milesi@auxologico.it (A.M.); l.pradotto@auxologico.it (L.G.P.); 3Department of Clinical Sciences and Community Health, University of Milan, 20122 Milan, Italy; cecilia.chighizola@unimi.it; 4U.O.C. Clinica Reumatologica Pediatrica, ASST G. Pini—CTO, 20122 Milan, Italy; 5Department of Neuroscience, University of Turin, 10124 Turin, Italy

**Keywords:** three-dimensional models, scaffold, hydrogel, fibroblasts, endothelial cells, dynamic culture, systemic sclerosis

## Abstract

Two-dimensional in vitro cultures have represented a milestone in biomedical and pharmacological research. However, they cannot replicate the architecture and interactions of in vivo tissues. Moreover, ethical issues regarding the use of animals have triggered strategies alternative to animal models. The development of three-dimensional (3D) models offers a relevant tool to investigate disease pathogenesis and treatment, modeling in vitro the in vivo environment. We aimed to develop a dynamic 3D in vitro model for culturing human endothelial cells (ECs) and skin fibroblasts, simulating the structure of the tissues mainly affected in systemic sclerosis (SSc), a prototypical autoimmune fibrotic vasculopathy. Dermal fibroblasts and umbilical vein ECs grown in scaffold or hydrogel, respectively, were housed in bioreactors under flow. Fibroblasts formed a tissue-like texture with the deposition of a new extracellular matrix (ECM) and ECs assembled tube-shaped structures with cell polarization. The fine-tuned dynamic modular system allowing 3D fibroblast/EC culture connection represents a valuable model of the in vivo interplay between the main players in fibrotic vasculopathy as SSc. This model can lead to a more accurate study of the disease’s pathogenesis, avoiding the use of animals, and to the development of novel therapies, possibly resulting in improved patient management.

## 1. Introduction

In vitro two-dimensional (2D) cell cultures and in vivo animal models have been fundamental in understanding human diseases and the biological effects of drugs. However, 2D cultures have limitations due to the lack of tissue-specific architecture, cell-to-cell interactions, and cell-to-matrix interactions [1,2,3,4]. On the other hand, ethical issues concerning animal welfare have led to the establishment of the principle of the three R’s: Replacement, Reduction, and Refinement [5,6,7,8], pushing scientists worldwide to replace animals with alternative experimental strategies. These limitations can be partially overcome, by the employment of three-dimensional (3D) in vitro models [1,2,9,10]. These 3D models can closely mimic the in vivo environment of native tissues [11,12]. Recent advances in cell biology, microfabrication techniques, and tissue engineering have made it possible to create different methods of assembling 3D cell cultures [1,2]. Cells may be encapsulated in scaffold-based structures that mimic the extracellular matrix (ECM) architecture or assembled in non-scaffold-based cellular aggregates known as spheroids [13]. These models have improved the evaluation and monitoring of cell morphology/functions and are being widely used in biomedical and pharmacological research [13]. Additionally, 3D systems are particularly useful for studying phenotypes poorly reproducible in conventional 2D cell cultures, such as angiogenesis [12,13,14]. A wide variety of 3D models have been developed, but their main limitation is the lack of standard approaches. Due to the complexity of these systems, several parameters (e.g., substrate, cell source, flow rate) need to be considered in order to set up culture conditions and identify the most suitable growth environment tightly mimicking the native tissue [15,16]. Bioreactors have become increasingly important in constructing in vitro tissue and organ models because of their ability to maintain cells and tissues in controlled physical conditions. The bioreactor cell culture chambers operate in a dynamic culture environment and can be used independently or inter-connectedly in modular multi-compartmental platforms simulating a multi-organ system. Despite the complexity of the starting setting, the use of bioreactors in 3D cultures allows the study of cell functions, interactions, and crosstalk between different tissues/organs under flow conditions, reproducing the mechanisms of in vivo perfusion [2,11,12,13,14,15,16,17]. Fibrotic vasculopathies are characterized by vascular damage together with the deposition of fibrotic tissue in organs and inflammation. The archetypal vasculopathy is systemic sclerosis (SSc), a chronic systemic autoimmune disease characterized by endothelial damage, defective vasculogenesis, impaired vessel repair, immune system dysregulation, and abnormal ECM deposition resulting in tissue fibrosis. SSc’s clinical manifestations are heterogeneous and may involve the skin and internal organs. Skin abnormalities occur early in the disease course. They include vascular alterations, such as nail fold capillary changes, digital ulcers due to ischemic injury, cutaneous microvascular dysfunction, and progressive fibrosis, which predicts early internal organ involvement. SSc’s unique feature is that vascular dysfunctions and disease-specific autoantibodies precede fibrosis [18,19,20,21]. Although the pathogenic mechanisms of this complex disease are not completely understood yet, fibroblasts and endothelial cells (ECs) are deemed key players. Using the classical 2D in vitro culture model, we have recently demonstrated that immune complexes (ICs) containing SSc-specific autoantibodies can induce a pro-inflammatory and pro-fibrotic phenotype in both ECs and fibroblasts, contributing to SSc pathogenesis [21,22]. Furthermore, a huge variety of ICs in SSc patients was recently reported to contribute to the disease through IC-induced inflammatory reactions [23].

To confirm and strengthen the pathogenic role of SSc-derived immune complexes [21,22], we aimed to develop scaffold- and hydrogel-based 3D models for growing fibroblasts and ECs, respectively, to reproduce the architecture of the tissues mainly affected in SSc. Furthermore, we fine-tuned a dynamic 3D culture setting to grow ECs and fibroblasts in controlled conditions by applying bioreactors to reproduce the in vivo micro-environmental features.

## 2. Results

### 2.1. Fine-Tuning of Optimal Cell Culture Conditions

We used the IVTech modular multi-compartmental system for the set-up of dynamic 3D cell cultures (Figure 1).

Initially, we evaluated the growth of human dermal fibroblasts and umbilical vein endothelial cells (HUVECs) in the live bioreactors (LBs) by performing 2D cultures in a LB1 in static conditions, followed by the application of a 100 μL/min flow. Both cell types reached confluence within 48 h and the monolayers remained intact until 8 days of culture (6 days in static conditions plus 2 days in dynamic conditions). These time and flow parameters were then applied to the 3D models.

### 2.2. Human Dermal Fibroblasts Can Colonize and Reshape Bone Scaffold

Histological and immunocytochemical analyses evidenced that dermal fibroblasts, grown for 6 days into the Xylyx Bio porcine TissueSpec^®^ Bone ECM scaffold in LB1 in static conditions, induced the bone tissue reshaping (Figure 2).

The DNA-labeling dyes propidium iodide (PI) and 4′,6-diamidino-2-phenylindole (DAPI) localized the nuclei, highlighting the cellular organization in the matrix. As shown in Figure 2A, the bone scaffold in the absence of fibroblasts (negative control) consisted of compact tissue with closely packed osteons surrounded by concentric rings (lamellae) of the matrix. After the addition of the cells and their scaffold colonization, the bone tissue was remodeled, acquiring the aspect of spongy bone with newly formed tissue stained in pink with hematoxylin and eosin (H&E). The Gomori’s Trichrome (GTR) staining further supported the tissue remodeling by fibroblasts, with newly formed collagen in red and bone tissue in green-blue. The expression of alpha-smooth muscle actin (αSMA, green) confirmed fibroblasts’ activity and the detection of collagen IV (red) indicated the secretion of new ECM (Figure 2B).

Despite these promising results, the consistency and hardness of the bone scaffold made it very difficult to obtain regular sections, preventing further processing steps, such as homogenization for molecular analyses. These technical issues discouraged us from applying the dynamic conditions to the 3D cultures. To overcome this critical point, we replaced the bone scaffold with the new porcine TissueSpec^®^ Skin ECM scaffold.

### 2.3. Human Dermal Fibroblasts Can Colonize Skin Scaffold and Develop a New Texture

Dermal fibroblasts were incubated in a skin scaffold housed in LB1 for 6 days in static conditions and then a flow (100 μL/min) was applied for 48 h. Histological and immunocytochemical analyses were performed in all the experimental conditions. The skin matrix was softer and easier to handle than the bone scaffold.

Figure 3 highlights the difference between the empty scaffold (acellular structure as the negative control) and scaffolds in the presence of fibroblasts cultured either in static or dynamic milieu.

Figure 3A shows that dermal fibroblasts in static conditions took up the whole scaffold with a random distribution, evidenced by both H&E and GTR (new ECM in red and dermal scaffold in dark blue). The presence of the cells was proven by nuclear staining with PI. When flow was applied (dynamic condition), the tissue texture underwent a structural change, with fibroblasts migrating to connect with each other and form a more organized tissue, with increased ECM deposition (increased red areas with H&E and GTR staining, which identified the dermal scaffold in dark blue). Cell fusion was evidenced by PI. The detection of collagen IV (red) and αSMA (green) in static conditions by immunofluorescence analysis (Figure 3B) further supported an initial deposition of a new ECM by the cells (blue nuclei) and confirmed its enhancement under flow conditions.

### 2.4. Endothelial Cells Can Develop Primitive Vascular Plexus in Lung ECM Hydrogel

Data in the literature [24,25] indicate that the 3D growth of ECs requires matrices providing a cellular microenvironment similar to the in vivo one, with high aqueous content and availability of suitable growth factors, suggesting hydrogel as the best solution.

We grew HUVECs in lung ECM hydrogel in static conditions for 6 days, followed by 48 h of culture under flow (100 μL/min). Figure 4A illustrates the histology results obtained with H&E staining in the different experimental settings.

The cell growth into the hydrogel was evidenced by PI nuclear staining (red). No cellular texture was observed in the control hydrogel without cells. Endothelial cells cultured in static conditions spread across the hydrogel, leading to the development of early blood islands and the formation of infant tube-shaped structures. After culture medium perfusion, we observed an improvement of the cellular organization, with the fusion of the blood islands and their remodeling into better-defined tubular structures ending with the formation of the first primitive vascular plexus. The histology data were confirmed by the evaluation of the expression of CD31, a specific endothelial surface marker (Figure 4B).

The vascular endothelium is made up of polarized cells and the structure, organization, and positioning of the Golgi apparatus are implicated in maintaining a polarized cell state [26]. We evaluated the expression and localization of the Golgi matrix protein 130 kDa (GM130), a structural element of the Golgi apparatus, in HUVECs grown in static and dynamic environments. As shown in Figure 5, in dynamic conditions, the Golgi apparatus (green) was clearly separated from the nuclei (blue) and oriented towards the front of the edge cell membrane, supporting the transition from non-polarized to polarized HUVECs in response to flow. No staining was observed in hydrogel without HUVECs (negative control).

## 3. Discussion

We set up two different 3D in vitro models using modular bioreactors for culturing human skin fibroblasts and macrovascular ECs in a dynamic environment that closely resembles the SSc in vivo milieu. SSc presents with a variety of clinical symptoms, all of which are caused by a common disease-specific pathologic cascade across multiple organs [27,28]. To date, the most reliable tool to predict the pattern of organ involvement and stratify patients with different severities and prognoses is provided by the fine specificity of SSc-associated autoantibodies [29], as supported by the most recent classification criteria [30]. For example, antibodies against DNA topoisomerase I or Th/To ribonucleoprotein are predictors of the development of interstitial lung disease (ILD) while antibodies against centromeric proteins are mainly associated with calcinosis and pulmonary arterial hypertension, together with a low risk of ILD. Furthermore, antibodies directed against RNA polymerase III provide one of the strongest risk factors for renal crisis and have been linked to severe cutaneous involvement [28,29]. The disease specificity and the role of SSc prognostic biomarkers have suggested the potential pathogenicity of these autoantibodies [31]. Our previous studies, showing that SSc-specific autoantibodies embedded in ICs can induce a pro-inflammatory and pro-fibrotic phenotype in fibroblasts and ECs in traditional 2D in vitro systems, are consistent with the pathogenic relevance of these autoantibodies [21,22]. However, 2D cell culture models do not allow the maintenance/reproduction of the tissue architecture and the direct interaction existing in vivo among fibroblasts and ECs, the main players in the development of SSc. On the other hand, the SSc murine models recently used to investigate disease pathogenesis [32,33,34] have some limitations: (i) most of them resemble only some of the SSc features, (ii) they do not display autoimmune responses, and (iii) they frequently have a local, rather than a systemic, disease. Furthermore, the ethical issues concerning animal welfare and the increasing importance acquired by the principle of the three R’s address researchers’ interest in alternative models [5].

Three-dimensional cell cultures offer a promising alternative way to model in vitro and in vivo environments, eventually bridging the gap between 2D cell culture and animal models [1,2]. Several technologies have been developed to enable cellular organization in well-defined 3D structures [12,13,14]. Of note, different cell types require different matrices that are similar to the tissue of origin [10]. Different 3D models of fibroblast culture have been developed to understand the crucial events leading to fibrosis, particularly in lung diseases (e.g., idiopathic pulmonary fibrosis, IPF) and cancer. These models use different techniques depending on the goals of the study, such as examining cell interactions, fibroblast migration, and differentiation, identifying molecular mechanisms involved in fibrosis or regeneration and drug discovery. Examples of these techniques include hydrogels, scaffolds, precision-cut lung slices (PCLS), organoids, and lung-on-chip models [35,36]. Employing hydrogels of different smoothness or soft scaffolds has allowed us to investigate the microenvironment mechanical dysfunctions in IPF and cancer. Controlling the scaffold porosity has influenced the phenotype of activated fibroblasts in 3D cultures, favoring the study of early and late stages of fibrosis and cancer [36]. Patient-derived PCLS have been used for reproducing IPF early fibrosis and for drug testing. Moreover, stem-cell-derived lung organoids have been useful in mimicking IPF physiopathologic conditions while lung-on-chip models have enabled the co-culture of fibroblasts, vascular and lung epithelial cells from IPF patients, or healthy donors to characterize the IPF phenotype [35]. Additionally, 3D dynamic platforms employing spheroids, microcarrier beads, or 3D round section microchannels have also been developed to closely reproduce the in vivo cell–blood interaction cross-talk between ECs and other cell types and EC functions [37,38,39].

In this study, we describe, for the first time, the setting up of independent 3D cultures of human dermal fibroblasts seeded in a scaffold and HUVECs in lung ECM hydrogel using IVTech bioreactors, both in static and dynamic conditions. The matrix selection was based on compatibility with our bioreactors. Initially, fibroblasts were cultivated in a porcine bone scaffold, which was deemed the most suitable option among the available ones at the beginning of this study. Although the cells penetrated the matrix and produced a new ECM, handling the substrate was extremely challenging. The availability of skin scaffolds not only made the processing easier but also offered a support structure much closer to the tissue from which fibroblasts were obtained. After the matrix colonization in static conditions, dermal fibroblasts improved their structural organization and increased the de novo synthesis of human connective ECM under flow conditions, as demonstrated by the αSMA/collagen IV-positive phenotype. These findings support the efficacy of the bioreactor chamber in providing a physiological environment for fibroblasts on a 3D skin scaffold.

Concurrently, we observed that HUVECs encapsulated within lung ECM hydrogel, which has a jelly-like texture and is enriched with Xylyx Bio proprietary growth factors, developed blood islands and formed capillary cord-like structures under flow conditions. Vasculogenesis is a dynamic process that involves cell–cell and cell–ECM interactions in the presence of growth factors and regulatory proteins. It leads to the formation of new blood vessels from angioblasts, giving rise to ECs [40,41,42]. Platelet endothelial cell adhesion molecule 1 (PECAM-1 or CD31) is one of the most sensitive and specific angiogenesis markers and is enriched at EC intercellular junctions. This molecule is involved in cell migration and enables the formation of new blood vessels through cell–cell adhesion [43,44]. The expression of CD31, particularly evident after perfusion, demonstrates that the proposed 3D culture model can favor the creation of an interconnecting network of ECs that mimics the early stages of blood vessel generation. ECs are the cells that line the inside of blood vessels. They are constantly exposed to blood flow that exerts shear stress on them. This shear stress plays a role in regulating EC migration, modifying their contacts with each other, and remodeling their cytoskeleton. These observations suggest that the formation of new blood vessels requires the polarization of these cells in response to flow, with the apical surface of the cell facing the fluid and the basal surface facing the ECM [45,46]. In dynamic conditions, we observed the migration of the Golgi apparatus in front or behind the nuclei depending on the cells’ position and flow direction [47,48]. These data further support the suitability of our 3D model combining the lung ECM hydrogel and bioreactor for reproducing vascular tissue in vitro.

3D scaffold-free and scaffold-based models of SSc have been implemented, mainly for drug development and screening [49,50]. Defective angiogenesis in SSc has recently been studied, employing a 3D microvessel-on-a-chip model [51]. The overall results indicate the IVTech dynamic multi-compartmental system as an evolution of the traditional 2D cultures and a valuable in vitro alternative to animal employment. By growing fibroblasts and ECs in three dimensions with appropriate matrices and exposing them to a fluidic environment mimicking blood flow, the system allows testing the effects of molecules like plasma proteins or drugs on vascular or connective tissue-like structures. The modular bioreactors offer the possibility of setting up a two-way connected culture of fibroblasts and ECs, representing a simplified model of the in vivo interplay between the major cellular players in the pathogenesis of SSc. This advanced model will help us deeply explore the pathogenic role of SSc-specific ICs, previously demonstrated in standard 2D in vitro cultures [21,22]. Moreover, the modularity of the IVTech platform reproducing the complexity of multi-tissue interactions might further expand the actual knowledge about the pathogenic mechanisms in SSc and more in general in fibrotic vasculopathies, also providing important clinical implications leading to the development of novel therapeutic strategies.

## 4. Materials and Methods

### 4.1. Modular Multi-Compartmental Cell Culture System

We used the system developed by IVTech (IVTech Srl, Ospedaletto, PI, Italy) to reproduce in vitro 3D models of endothelium and connective tissue in a dynamic environment simulating the blood flow. The system allows 2D and 3D cell culturing both in static and dynamic conditions. It provides two different transparent 24-well plate-like culture chambers: LiveBox (LB) Type 1 and Type 2. LB1 is equipped with a removable glass slide supporting the cell monolayer or 3D constructs and a flow inlet and outlet for the perfusion of culture media. LB2 allows the modeling of physiological barriers in vitro through a porous selective membrane (specific for each cell type) housed in a removable holder. It also has two flow inlets and outlets. For the purpose of setting up the best culture conditions, we employed only LB1 for culturing both fibroblasts and HUVECs. The bioreactors are designed for interconnected dynamic cell cultures thanks to their modularity that mimics the cross-talk between tissues, leading to a multi-organ simulation. A clamp system provided with the LBs assures the watertight closure of the chambers. A peristaltic pump (LiveFlow) connected to bioreactors and reservoirs applies an adjustable flow rate (range 100–450 μL/min) and enables cell–cell interactions through soluble molecules/proteins as they occur in vivo. Two removable pumping heads drive two independent circuits. Three-way valves positioned along the fluidic circuit allow the administration of drugs/modulators to the cells or the collection of conditioned media from the cultures at different sites of the system. Figure 1 shows the IVTech (IVTech Srl, Ospedaletto, PI, Italy) system assembled as required by the experimental plan.

### 4.2. Primary Cells

#### 4.2.1. Human Healthy Skin Fibroblasts

Human dermal fibroblasts were purchased from Promocell (Heidelberg, Germany) and cultured in D-MEM medium (Gibco-Life Technologies, Groningen, The Netherlands) supplemented with 10% heat-inactivated Fetal Bovine Serum (FBS, PAA-GE Healthcare, Buckinghamshire, UK), 2 mM glutamine (Sigma-Aldrich, St. Louis, MO, USA), and 100 μg/mL penicillin-streptomycin (Sigma-Aldrich, St. Louis, MO, USA) and maintained at 37 °C in a 5% CO_2_-humidified incubator until confluence. Fibroblasts were then passaged using trypsin/EDTA (ThermoFisher Scientific, Waltham, MA, USA), grew up to the 8th passage, and 400,000 cells in 20 μL complete medium were seeded.

#### 4.2.2. Human Macrovascular Endothelial Cells

HUVECs were purchased from Promocell (Heidelberg, Germany) and grown in the complete E199 medium (EuroClone S.p.A., Milan, Italy) supplemented with 20% heat-inactivated FBS (PAA Laboratories-GE Healthcare, Buckinghamshire, UK), 1% L-glutamine (Sigma-Aldrich, St. Louis, MO, USA), 100 U/mL penicillin-streptomycin (Sigma-Aldrich, St. Louis, MO, USA), and 250 ng/mL Amphotericin B (ThermoFisher Scientific, Waltham, MA, USA) in a humidified incubator at 37 °C and 5% CO_2_. Confluent cells were passaged with 0.05% trypsin/EDTA (ThermoFisher Scientific, Waltham, MA, USA) and 500,000 cells in 20 μL of the complete medium were seeded.

#### 4.2.3. 3D Cell Cultures

We selected a scaffold and a hydrogel to grow fibroblasts and HUVECs, respectively, taking into account the tissues of origin of the cells of interest.

Initially, we grew dermal fibroblasts into a porcine bone scaffold (Bone TissueSpec^®^ Scaffold, Xylyx Bio, Brooklyn, NY, USA); however, we found it challenging to handle. So, we switched to a skin scaffold of the same origin (Skin ECM Scaffold, Xylyx Bio, Brooklyn, NY, USA). Both are custom acellular matrix substrates. They retain the natural 3D structure, biomechanics, and topography of native tissues. Tissue-specific features and architecture peculiar to these products support fibroblast–ECM and fibroblast–fibroblast interactions, favoring cell migration, organization, and integration to form 3D constructs. Dermal fibroblasts were grown in the scaffold according to the manufacturer’s instructions. After a rehydration step in the complete medium, we spread 20 μL of cellular solution (400,000 fibroblasts) on the top of the scaffold. Then, we housed it in LB1 and incubated it at 37 °C in a 5% CO_2_ humidified environment for 45 min to allow cellular penetration. Then, we added 1.5 mL of complete medium to the system. We maintained two LB1s in static conditions for 6 days; then, we exposed one of them to a 100 μL/min flow for a further 48 h. At the end of each defined culture period, we removed the scaffolds, washed them in PBS, fixed them in 4% Paraformaldehyde (PFA, Bio-Optica, Milan, Italy), and processed them for histological analyses and immunochemistry. As a negative control, scaffolds without cells were cultured in the same experimental conditions.

Studies have shown that the most suitable microenvironment for the 3D growth of ECs should be highly aqueous and enriched with fibroblast-derived angiogenic macromolecules and growth factors [52]. As such, we grew HUVECs in the Xylyx Bio porcine lung ECM hydrogel (TissueSpec^®^ ECM Hydrogel Kit, Xylyx Bio, Brooklyn, NY, USA). This is a versatile extracellular matrix composed of collagens and other ECM molecules of lung-specific origin, which provides the cells with an easy-to-use, soft, physiologic substrate favoring vascular morphogenesis in the 3D environment.

HUVECs (500,000/20 μL) were mixed with the ECM hydrogel prepared according to the manufacturer’s protocol (final concentration 6 mg/mL) and the mixture was seeded in two LB1s. The bioreactors were incubated at 37 °C in a humidified 5% CO_2_ environment for 45 min to achieve gelation and encapsulation of the cells within the hydrogel. Afterward, 1.5 mL of complete medium was added and the two LB1s were maintained for 6 days in an incubator in static conditions to allow cell growth and organization. Flow was then applied at 100 μL/min to one of them by a peristaltic pump for a further 48 h. At the end of the culture period, the hydrogel was removed from the bioreactors, fixed in PFA 4%, and processed for histology and immunochemistry. Hydrogels without cells were cultured in parallel as negative controls. The different experimental conditions are depicted in Figure 6.

### 4.3. Histological Analyses

After fixation and paraffin embedding, we cut the scaffolds (3D fibroblasts) into 5–10 μm thick slices using a microtome (Leika, Wetzlar, Germany) and stuck them onto glass slides coated with polylysine (Superfrost Plus, ThermoFisher Scientific, Waltham, MA, USA). The sections were cleared in xylene substitute (Bioclear, Bio-Optica, Milan, Italy) and dehydrated by immersion in ethanol (Sigma-Aldrich, St. Louis, MO, USA; from 100% to 50% serial dilutions).

We embedded the fixed hydrogels (3D HUVECs) in an Optimal Cutting Temperature (OCT) compound, froze them in liquid nitrogen, and sectioned them at 10 μm thickness using a cryostat (Leica Biosystems, Weitzlar, Germany) at −25 °C. We stuck the sections onto glass slides and stained them: (i) with H&E (Bio-Optica, Milan, Italy) to evaluate tissue morphology and structure and (ii) with GTR to evidence scaffold remodeling and new ECM deposition. We analyzed the 3D growth/organization of the cells both in static and dynamic environments by light microscopy (Leica FC7000 T, Leica Biosystems, Weitzlar, Germany).

### 4.4. Immunocytochemical Analysis

To demonstrate the de-novo synthesis of ECM components by fibroblasts and an ongoing vasculogenesis supported by HUVECs in our 3D cultures, we performed marker-specific immunofluorescence staining on scaffold and hydrogel sections prepared as described above. We incubated dewaxed scaffold sections for one hour with the primary mouse monoclonal antibody directed to human αSMA (1:100, Sigma Aldrich, St. Louis, MO, USA) together with the primary rabbit polyclonal anti-human collagen IV (1:1500, Cederlane, Burlington, ON, Canada) in PBS/1% BSA. After that, we added a secondary antibody (Alexa Fluor 488-conjugated anti-mouse or Alexa Fluor 555-conjugated anti-rabbit IgG, ThermoFisher Scientific, Waltham, MA, USA) for 30 min at room temperature (RT). Nuclei were counter-stained with DAPI (GeneTex, Irvine, CA, USA), a blue-fluorescent dye that binds to AT-rich regions of dsDNA. Finally, we mounted the slides using the ProLong^TM^ Gold Antifade Mountant (ThermoFisher Scientific, Waltham, MA, USA), specifically designed to protect fluorescent dyes from fading.

We evaluated the organization of HUVECs to form primitive blood vessels in the hydrogels as the expression of CD31, constitutively expressed on the surface of EC, by a mouse anti-human CD31 monoclonal antibody (1:50, Sino Biological Inc., Eschborn, Germany). To investigate EC polarization, a rabbit monoclonal antibody GM130 specific to the human Golgi apparatus (1:2000, Proteintech, Manchester, UK) was used. Briefly, after washing twice with PBS, the sections were permeabilized in PBS plus 0.1% Triton X100 (PBS-T, Sigma-Aldrich, St. Louis, MO, USA) and blocked in 2% BSA for one hour. The mouse anti-human CD31 or rabbit anti-human GM130 antibodies were then added for one hour, washed thrice, and incubated with the AlexaFluor 488-conjugated goat anti-mouse or anti-rabbit IgG secondary antibody (1:500 PBS-T/1% BSA) for one hour. Finally, after further washing, the samples were stained with DAPI for 10 min and the slides were mounted using the ProLong^TM^ Gold Antifade Mountant.

To better visualize cell distribution, we washed the dewaxed scaffolds and frozen hydrogel sections for 5 min and stained them with PI (ThermoFisher Scientific, Waltham, MA, USA) (33 μg/mL final dilution, 30 min). After washing twice, the slides were mounted using ProLong^TM^ Gold Antifade Mountant.

Images were acquired using a confocal laser scanning microscope (Nikon Eclipse Ti, Nikon, Tokio, Japan) and analyzed with the Nikon universal software platform NIS-Elements Advance Research (AR) version 5.02.0064-BIT.

## 5. Conclusions

The dynamic 3D in vitro culture model of endothelial cells and fibroblasts is a promising approach to studying systemic sclerosis, a fibrotic vasculopathy. This model enables the investigation of the mechanisms involved in the disease by reproducing the tissue architecture and interconnections in vitro while avoiding the ethical issues associated with animal handling.

Moreover, this multi-compartmental culture system can be adapted to different physiopathologic conditions that involve endothelial cells and fibroblasts. This approach can help to deepen the understanding of the pathogenesis of diseases beyond systemic sclerosis and to evaluate the effects of new drugs that target different components of the identified pathogenic pathways, enabling researchers to develop novel therapies and improve patient care.

## Figures and Tables

**Figure 1 ijms-25-02780-f001:**
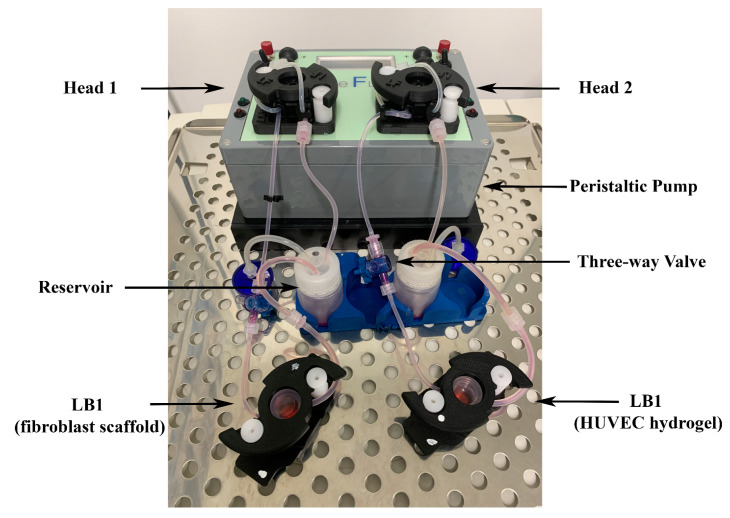
Modular multi-compartmental cell culture system. A Live Bioreactor 1 (LB1) houses human dermal fibroblasts and human umbilical vein endothelial cells (HUVECs) grown in the porcine scaffold or porcine lung hydrogel on a glass slide. A peristaltic pump equipped with two heads (Head 1 and Head 2) is connected to each LB1 and enables the reproduction of the blood flow by applying a modifiable flow rate. Three-way valves are placed in appropriate positions along the circuit to allow the injection of different stimuli/mediators and/or sampling of culture supernatants for further analyses.

**Figure 2 ijms-25-02780-f002:**
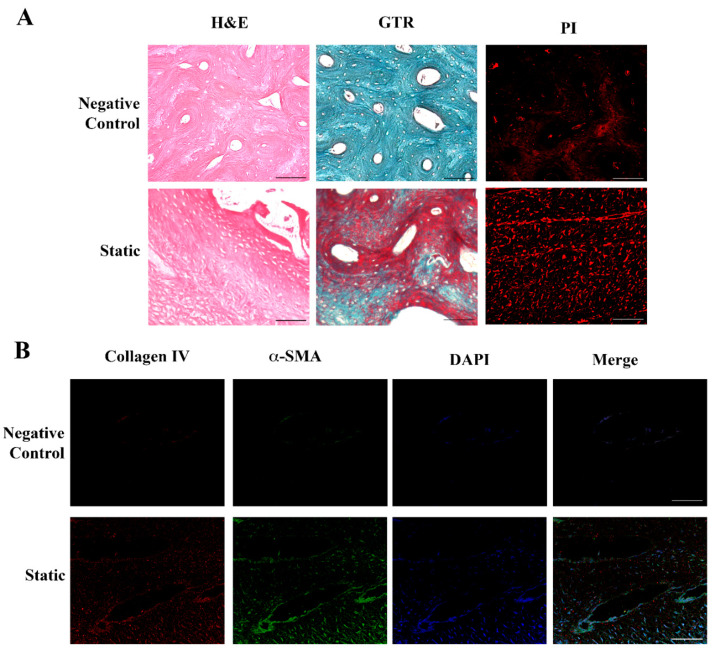
Histological and immunohistochemical analysis of fibroblasts seeded in the bone scaffold. Panel (**A**) (Upper): porcine bone scaffold in the absence of cells (negative control). Hematoxylin and Eosin (H&E) and Gomori’s Trichrome (GTR) staining evidence the compact structure typical of the bone tissue, with closely packed osteons surrounded by concentric rings of matrix. Lower: human dermal fibroblasts cultured in the porcine bone scaffold in static conditions for 6 days. The spongy aspect of the bone, together with the deposition of a new matrix highlighted by the H&E and GTR staining, indicates tissue remodeling. Propidium iodide (PI) identifies the nuclei, showing the cell distribution. Original magnification 20×. Panel (**B**) (Upper): porcine bone scaffold in the absence of cells (negative control). Lower: immunofluorescence analysis of human dermal fibroblasts grown in the porcine bone scaffold in static conditions. Collagen IV (red) and α-smooth muscle actin (αSMA, green) merged, indicating an initial production of a new extracellular matrix (ECM) by active fibroblasts. Nuclei were stained with 4′,6-diamidino-2-phenylindole (DAPI, blue) to identify cells. Original magnification 20×. Scale bars correspond to 100 μm in all the images.

**Figure 3 ijms-25-02780-f003:**
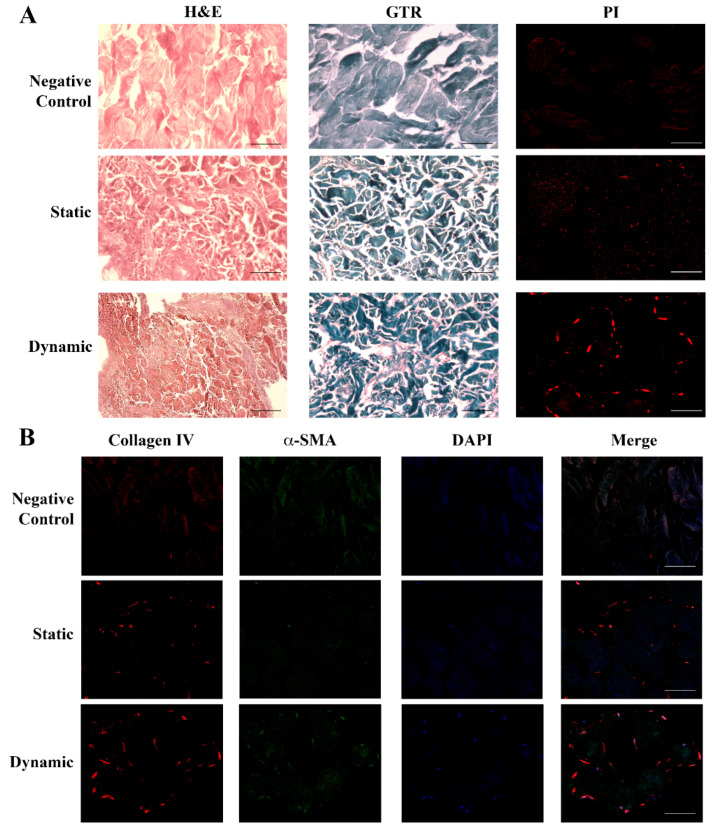
Histological and immunohistochemical analysis of fibroblasts seeded in the dermal scaffold. Panel (**A**) (Upper): negative control showing the acellular structure of the porcine dermal scaffold without fibroblasts either with H&E or GTR stainings. (Central): sections of the porcine dermal scaffold in the presence of human dermal fibroblasts cultured for 6 days in static conditions and stained with H&E or GTR showed an initial tissue remodeling with the random cellular distribution. (Lower): porcine dermal scaffold in the presence of human dermal fibroblasts cultured in static conditions for 6 days followed by 48 h under flow. Images showed a change in the tissue texture and cellular organization together with new collagen deposition. Nuclei were stained with PI. Original magnification 20×. Panel (**B**) (Upper): negative control. The initial deposition of collagen IV (red) and expression of α-SMA (green) visible in static conditions (Central) increased underflow, leading to the formation of a more organized tissue structure and consistent ECM deposition (Lower): dynamic. Nuclei were stained with DAPI (blue). Original magnification 20×. Scale bars correspond to 100 μm in all the images.

**Figure 4 ijms-25-02780-f004:**
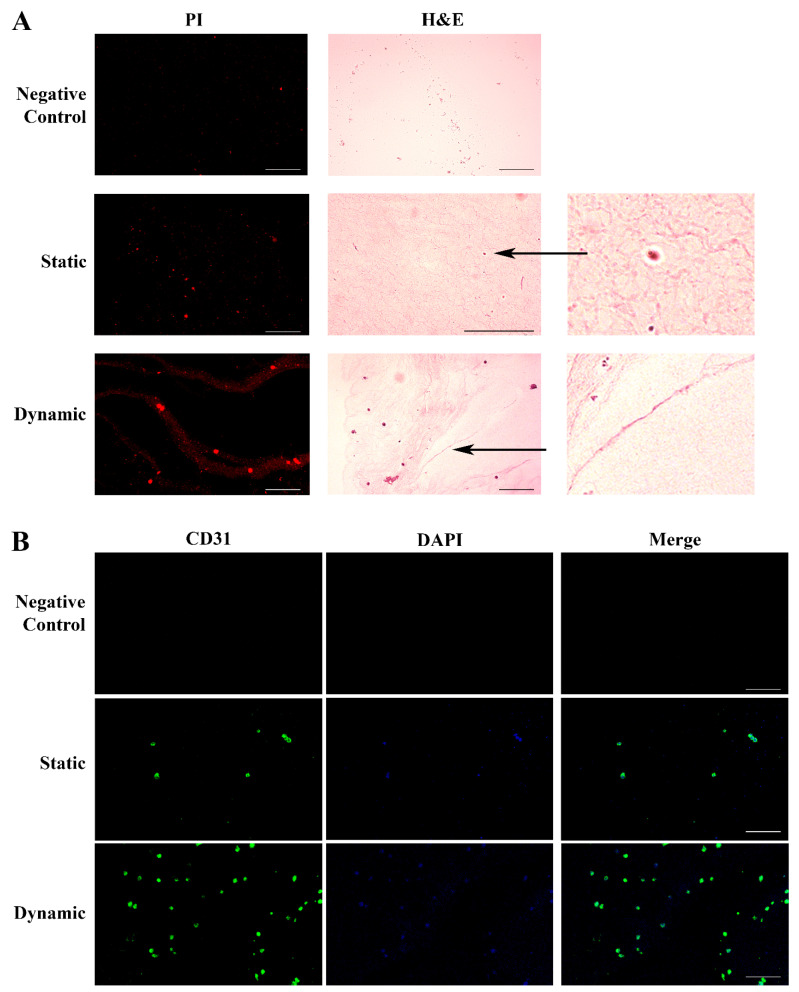
Histological and immunohistochemical analysis of HUVECs grown in the ECM lung hydrogel. Panel (**A**). H&E staining of ECM hydrogel in the absence of cells (Upper: Control); HUVECs spread across the hydrogel in static conditions forming early blood islands (Central: Static); HUVECs cultured in the hydrogel under flow, displaying a more organized texture with a fusion of the blood islands and their remodeling into tubular structures forming a primitive vascular plexus (Lower: Dynamic). Staining of the nuclei with PI (red) identified the cells. Original magnification 20×, except for H&E static image 40×. The two additional panels represent specific details of blood islands and vascular plexus indicated by the arrows (4×). Panel (**B**). HUVECs grown in the hydrogel were identified by CD31 expression (green) on the cell membrane in static conditions (Central), which strongly increased under flow, showing a better-organized architecture (Lower). Nuclei were stained with DAPI (blue). The hydrogel in the absence of HUVECs was negative for both stainings (Upper). Original magnification 20×. Scale bars correspond to 100 μm in all the images.

**Figure 5 ijms-25-02780-f005:**
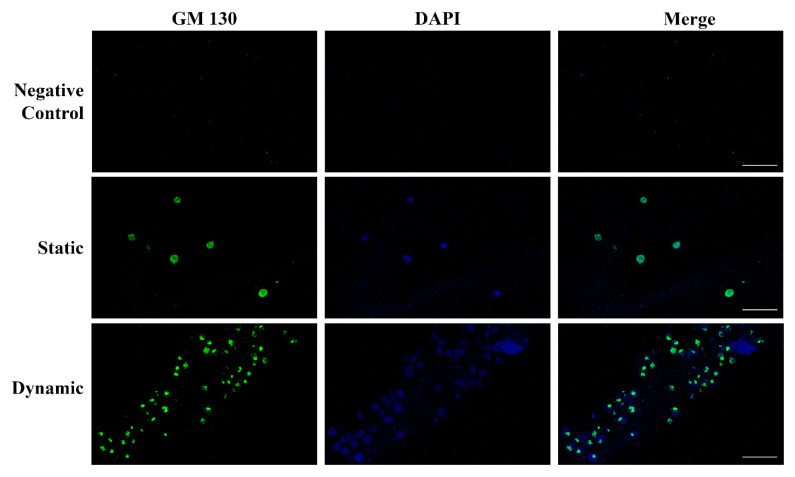
Polarization of HUVECs grown in the ECM lung hydrogel. Upper: negative control hydrogel does not show any staining, confirming the absence of ECs. Central: in static conditions, the Golgi matrix protein GM130 (green) is homogeneously distributed into the cell cytoplasm referring to the nucleus (blue). Lower: the HUVEC polarization is clearly evidenced in dynamic conditions by the localization of the GM130 (green) behind or in front of the nuclei (blue). Original magnification 40×. Scale bars correspond to 50 μm in all the images.

**Figure 6 ijms-25-02780-f006:**
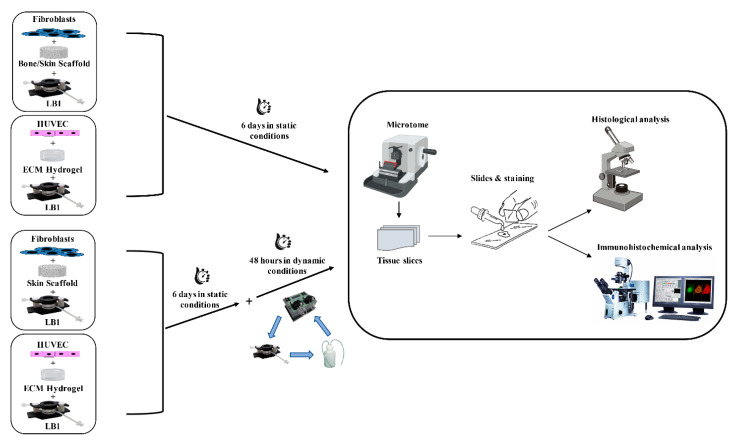
Flowchart of the 3D cell cultures. Fibroblasts were seeded in bone or skin scaffold and HUVECs were in the ECM hydrogel. The 3D cultures were maintained in LB1 for 6 days in static conditions. Concurrently, the same cell types were housed in LB1 in skin scaffold or ECM hydrogel, respectively, for 6 days in a static milieu and then exposed to 100 μL/min flow for a further 48 h. At the end of the culture period, samples were processed for histological or immunohistochemical analyses.

## Data Availability

The data presented in this study are available on request from the corresponding author.

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
