# Peer review of "Fibroblasts and Endothelial Cells in Three-Dimensional Models: A New Tool for Addressing the Pathogenesis of Systemic Sclerosis as a Prototype of Fibrotic Vasculopathies"

_ijms, 2024, doi:10.3390/ijms25052780_

Round 1

Reviewer 1 Report

Comments and Suggestions for Authors

The authors propose a dynamic 3D model for culturing endothelial cells and fibroblasts with subsequent study of the pathogenesis of systemic sclerosis. However, the authors showed only the possibility of using the dynamic milieu to create three-dimensional monocellular structures. It is not entirely clear how this is directly related to systemic sclerosis, it is rather a perspective. Probably, it would have been worthwhile to place the emphasis in Introduction and Discussion differently and focus on the result that was obtained. After proper corrections the paper can be published. 

The comments include requirements and recommendations.

1) Introduction

‘In this study, we aimed to develop scaffold- and hydrogel-based 3D models for growing fibroblasts and ECs, respectively, to reproduce the architecture of the tissues mainly affected in SSc’ is the purpose of the study. However, the last paragraph of the introduction does not serve the purpose, but is formulated as a conclusion. Reword, delete or move it.

2) 2.2. Human dermal fibroblasts can colonize and reshape bone scaffold

‘Histological and immunocytochemical analyses evidenced that dermal fibroblasts, grown for 6 days into the Xylyx Bio porcine TissueSpec® Bone ECM scaffold in LB1 in static conditions, induced the bone tissue reshaping’: Probably, dynamic conditions were not applied due to the complexity of working in bone scaffold. Write explicitly why you did not conduct the experiment in a dynamic milieu.

3) Figure 2

‘GTR staining indicate a tissue remodeling’: Write in the description of the figure what is stained red and what is blue.

4) Figure 3

‘GTR stainings’: Write in the description of the figure what is stained red and what is blue.

It is not indicated for how long the cells were cultured under static conditions before staining (panel A, central).

5) Discussion

The second paragraph stands out from the overall narrative and looks like it was added just to justify the title. The second paragraph better fits the Introduction.

The text of the Discussion needs to be worked on; the text does not look cohesive.

6) 4.3.D cell cultures

‘We maintained the scaffold for 6 days in static conditions and then exposed it to 100 mcl/min flow for further 48 hours’: You did not describe the static conditions. Or did you not conduct the experiment for dynamic and static conditions in parallel? If this is the case, then explicitly indicate that the staining was performed on the same structures before dynamic conditions. It may be worthwhile to outline the experimental design in diagrammatic form.

7) References

‘49.’: Extra list item.

Author Response

We would like to thank the referees for their constructive and insightful comments, which have helped us to improve our manuscript. Below, we have provided a point-by-point response to each of the comments raised by the Reviewers. We have marked in red the changes made in the text to address these points.

Peer-Reviewer 1

Comments 1: Introduction ‘In this study, we aimed to develop scaffold- and hydrogel-based 3D models for growing fibroblasts and ECs, respectively, to reproduce the architecture of the tissues mainly affected in SSc’ is the purpose of the study. However, the last paragraph of the introduction does not serve the purpose, but is formulated as a conclusion. Reword, delete or move it.

Response 1: We thank the Referee for his/her constructive comment. We have reworded the text [page 2, paragraph 2, line 33] and deleted the last paragraph of the introduction as suggested [page 2, paragraph 3, lines 39-43].

Comments 2: 2.2. Human dermal fibroblasts can colonize and reshape bone scaffold.

‘Histological and immunocytochemical analyses evidenced that dermal fibroblasts, grown for 6 days into the Xylyx Bio porcine TissueSpec® Bone ECM scaffold in LB1 in static conditions, induced the bone tissue reshaping’. Probably, dynamic conditions were not applied due to the complexity of working in bone scaffold. Write explicitly why you did not conduct the experiment in a dynamic milieu.

Response 2: As correctly understood by this Reviewer, the difficulties encountered in processing the bone scaffold discouraged us from applying the dynamic conditions to the fibroblast 3D cultures.  We have modified the text accordingly [page 5, paragraph 2, lines 11-12].

Comments 3: Figure 2 ‘GTR staining indicate a tissue remodeling’: Write in the description of the figure what is stained red and what is blue.

Response 3: In the description of Figure 2 we have indicated that newly formed collagen is colored in red and bone tissue in green-blue, as suggested [page 5, paragraph 1, line 6].

Comments 4: Figure 3 ‘GTR stainings’: Write in the description of the figure what is stained red and what is blue. It is not indicated for how long the cells were cultured under static conditions before staining (panel A, central).

Response 4: As suggested by the Referee, we have detailed that new ECM is stained in red and dermal scaffold in dark blue in the description of Figure 3 [page 7, paragraph 1, lines 4-5 and 8-9].

We have added the time course of static conditions in the legend of Figure 3, Panel A, Central [page 6, paragraph Figure 3 legend, line 7].

Comments 5: Discussion The second paragraph stands out from the overall narrative and looks like it was added just to justify the title. The second paragraph better fits the Introduction. The text of the Discussion needs to be worked on; the text does not look cohesive.

Response 5: We thank this Reviewer for his/her constructive criticism. We have rephrased the discussion to make the text more cohesive [page 10].

Comments 6: 4.2.3. 3D cell cultures ‘We maintained the scaffold for 6 days in static conditions and then exposed it to 100 mcl/min flow for further 48 hours’: You did not describe the static conditions. Or did you not conduct the experiment for dynamic and static conditions in parallel? If this is the case, then explicitly indicate that the staining was performed on the same structures before dynamic conditions. It may be worthwhile to outline the experimental design in diagrammatic form.

Response 6: We have detailed the experimental procedure in the text and added the flowchart as the Figure 6, as requested [pages 12, paragraph 2, lines 49-50; page 13, line 1; page 13, paragraph 2, lines 21-22 and Figure 6, lines 23-28].

Comments 7: References 49.’: Extra list item.

Response 7: In the previous version of the manuscript the Ref. 49 was a typo. However, in the revised text we have added new references (ref. 27-30) and updated the Reference list accordingly [page 16, paragraph References, lines 24-32].

Reviewer 2 Report

Comments and Suggestions for Authors

Thank you for allowing me to review this interesting paper on 3D modelling of fibroblasts and endothelial cells.

The paper is well written and reads well.

In particularly the figures/micrographs are are very instructive and well commented on. The authors make a clear message the 3D model could improve understanding of systemic sclerosis and hence may be of clinical relevance in novel human drug development and treatment regimens. 

As is I have no further suggestion/comments.

Author Response

We would like to thank the referees for their constructive and insightful comments, which have helped us to improve our manuscript. Below, we have provided a point-by-point response to each of the comments raised by the Reviewers. We have marked in red the changes made in the text to address these points.

Peer-Reviewer 2

Comments 1: Thank you for allowing me to review this interesting paper on 3D modelling of fibroblasts and endothelial cells.

The paper is well written and reads well.

In particularly the figures/micrographs are are very instructive and well commented on. The authors make a clear message the 3D model could improve understanding of systemic sclerosis and hence may be of clinical relevance in novel human drug development and treatment regimens. 

As is I have no further suggestion/comments.

Response 1: We are very grateful to this Reviewer for his/her flattering comments.

Reviewer 3 Report

Comments and Suggestions for Authors

The manuscript entitled " Fibroblasts and endothelial cells in three-dimensional models: a new tool for addressing the pathogenesis of Systemic Sclerosis as a prototype of fibrotic vasculopathies" has very interesting idea.

I have several recommendations to the authors:

1. "The archetypal vasculopathy is Systemic sclerosis (SSc), a chronic systemic autoimmune disease whose unique feature is that disease-specific autoantibodies and vascular dysfunctions precede fibrosis [18-20]. Although the pathogenic mechanisms of this complex disease are not completely understood yet, fibroblasts and endothelial cells are deemed as key players." It will be better if the authors add between this two sentences additional information about the vascular changes and digital ulcerations.

2. Add detailed legend of the abbreviations in fig. 2

3. The results part is well described.

4. "SSc is a prototype of fibrotic vasculopathies [18] and we have previously shown that SSc-specific autoantibodies embedded in ICs can induce a pro-inflammatory and pro-fibrotic phenotype in fibroblasts and ECs in traditional 2D in vitro systems"  - Authors can include additional information about pulmonary fibrosis and ILD and their relashionship with the antibodies in patients with high disease activity.

5. "Examples of these techniques include hydrogels, scaffolds, precision-cut lung slices, organoids, and lung-on-chip models [31,32]" - Include short summary of this results to confirm your statement. 

6. The materials and methods part is well-described.

7. It will be better if the authors remove the abbreviations from the conclusion.

8. Recheck the references. Number 49 is missing.

In conclusion, the manuscript presents  very innovative idea and it will be very interesting to the readers.

Comments on the Quality of English Language

The English language is on sufficient level 

Author Response

We would like to thank the referees for their constructive and insightful comments, which have helped us to improve our manuscript. Below, we have provided a point-by-point response to each of the comments raised by the Reviewers. We have marked in red the changes made in the text to address these points.

Peer-Reviewer 3

Comments 1: "The archetypal vasculopathy is Systemic sclerosis (SSc), a chronic systemic autoimmune disease whose unique feature is that disease-specific autoantibodies and vascular dysfunctions precede fibrosis [18-20]. Although the pathogenic mechanisms of this complex disease are not completely understood yet, fibroblasts and endothelial cells are deemed as key players." It will be better if the authors add between this two sentences additional information about the vascular changes and digital ulcerations.

Response 1: We thank the Referee for his/her suggestion. We have put in the text additional information about vascular disfunctions, fibrosis and clinical manifestations in SSc [page 2, paragraph 1, lines 17-25].

Comments 2: Add detailed legend of the abbreviations in fig. 2

Response 2: We have detailed the legend of Figure 2 as requested [page 4, paragraph Figure 2 legend, lines 10-12].

Comments 3: The results part is well described.

Response 3: We thank the Reviewer for his/her positive comment.

Comments 4: "SSc is a prototype of fibrotic vasculopathies [18] and we have previously shown that SSc-specific autoantibodies embedded in ICs can induce a pro-inflammatory and pro-fibrotic phenotype in fibroblasts and ECs in traditional 2D in vitro systems" - Authors can include additional information about pulmonary fibrosis and ILD and their relashionship with the antibodies in patients with high disease activity.]

Response 4: We thank the Reviewer for this helpful suggestion., We have modified the text accordingly, describing the relationship of the SSc-associated autoantibodies and disease severity in the Discussion section [page 10, paragraph 1, lines 5-15].

Comments 5: "Examples of these techniques include hydrogels, scaffolds, precision-cut lung slices, organoids, and lung-on-chip models [31,32]" - Include short summary of this results to confirm your statement.

Response 5:  As suggested by the Reviewer, we have added some details about the different 3D culture techniques in the Discussion section [page 10, paragraph 2, lines 37-47].

Comments 6: The materials and methods part is well-described.

Response 6: We thank the Reviewer for his/her positive comment.

Comments 7: It will be better if the authors remove the abbreviations from the conclusion.

Response 7: We have removed the abbreviations, as suggested [pages 14-15, paragraphs 1-2].

Comments 8: Recheck the references. Number 49 is missing.

Response 8: In the previous version of the manuscript the Ref. 49 was a typo. However, in the revised text we have added new references (ref. 27-30) and updated the Reference list accordingly [page 16, paragraph References, lines 24-32].